# Postural Sensorimotor Control on Anorectal Pressures and Pelvic Floor Muscle Tone and Strength: Effects of a Single 5P^®^ LOGSURF Session. A Cross-Sectional Preliminary Study

**DOI:** 10.3390/ijerph18073708

**Published:** 2021-04-02

**Authors:** Laura Fuentes-Aparicio, Beatriz Arranz-Martín, Beatriz Navarro-Brazález, Javier Bailón-Cerezo, Beatriz Sánchez-Sánchez, María Torres-Lacomba

**Affiliations:** 1Physiotherapy in Motion, Multi Speciality Research Group (PTinMOTION), Department of Physiotherapy, University of Valencia, 46010 Valencia, Spain; laura.fuentes@uv.es; 2Physiotherapy in Women’s Health (FPSM) Research Group, Physiotherapy Department, Faculty of Medicine and Health Sciences, University of Alcalá, 28801 Madrid, Spain; b.navarro@uah.es (B.N.-B.); javier.bailon@lasallecampus.es (J.B.-C.); beatriz.sanchez@uah.es (B.S.-S.); maria.torres@uah.es (M.T.-L.); 3Department of Physical Therapy, Centro Superior de Estudios Universitarios La Salle, Universidad Autónoma de Madrid, 28023 Madrid, Spain

**Keywords:** postural control, postural balance, anorectal pressures, pelvic floor muscle strength, pelvic floor muscle tone, pelvic floor disorders

## Abstract

Pelvic floor dysfunction (PFD) is a functional condition present most frequently in women. Despite pelvic floor muscle training being considered by the International Continence Society (ICS) as the first-line treatment in uncomplicated urinary incontinence, other more comprehensive postural methods as 5P^®^ LOGSURF have emerged. This preliminary cross-sectional study explores the effects of a single 5P^®^ LOGSURF session on pelvic floor muscle (PFM) tone and strength (MVC), resting anal tone, intrarectal pressure, and deep abdominal muscles activation. Thirty women were included (11 without PFD and 19 with PFD). Primary outcome measures were PFM tone, PFM MVC and resting anal tone and secondary measures outcomes were intrarectal pressure and deep abdominal activation. All outcome measures were collected before, throughout and after a single 30′ 5P^®^ LOGSURF session. The findings from this study suggest that PFM tone (PFD group: *p* = 0.09, *d* = 0.72; non-PFD group: *p* = 0.003, *d* = 0.49) and PFM MVC (PFD group: *p* = 0.016; non-PFD group: *p* = 0.005) decreased in both groups after a single 5P^®^ LOGSURF session, with a medium effect size for women with PFD. Contrarily, deep abdominal muscle MVC increased (PFD group: *p* < 0.001; non-PFD group: *p* = 0.03). Intrarectal pressure and resting anal tone decreased in both groups throughout the session. These results suggest that 5P^®^ LOGSURF method may be interesting if is performed by women with mild symptoms of PFD or healthy women to achieve a decrease in PFM tone in women who manifested pain to intracavitary techniques or practices. Further research with higher sample sizes and long-term are necessary for generalizing.

## 1. Introduction

Pelvic floor dysfunction (PFD) is a functional condition present most frequently in women that is associated with a significant reduction in their quality of life [1]. PFD include a wide variety of clinical conditions such as urinary incontinence (UI), anal incontinence, pelvic organ prolapses, lower urinary tract emptying and perception disturbances, defecatory dysfunctions, sexual disorders, and a variety of chronic pain syndromes of the perineal area. Of all of them, UI is the most prevalent (up to 40%). Pelvic floor muscle training (PFMT) is considered by the International Continence Society the first-line treatment with a Grade A recommendation in uncomplicated UI, due to its efficacy, simplicity, low cost, and absence associated adverse effects [2]. PFMT improves symptoms of UI in the short term, but success rates decline during follow-up. Although this long-term declined efficacy appears to be due to lack of adherence to exercise [3], other more comprehensive alternatives to PFMT have emerged. These propose a less analytical training, that is, less focused on the pelvic floor muscles (PFM), and more on static and dynamic postural control and sensorimotor control of the abdominopelvic cavity and the lower extremities [4,5,6,7,8,9,10,11,12,13].

These global treatments are supported by diverse findings that show differences in motor control during coughing and voluntary contraction of the PFM [14,15,16,17], differences in postural balance while standing [18,19,20], a decreased functional walking capacity in women with UI compared to continent women [19], the relationship between the loss of lumbar lordosis with PFD [21], and the relationship between decreased foot flexion and UI [22].

Different studies in healthy women suggest the functional relationship between PFM, deep trunk erectors, and deep abdominal muscles [23,24,25], as well as abdominopelvic synergies [26]. Thereby, it is likely that PFD can affect lumbopelvic motor control, and that any deficit in motor control of the spine and pelvis may have consequences on continence [27]. Likewise, the literature describes synergies between the gluteus maximus muscle and the PFM that would explain the contribution of PFM in the upright and vertical posture, as well as with the respiratory muscles, mainly with the external intercostal muscles [23,28]. An anticipated postural adjustment to movements of the upper and lower limbs has also been described regarding PFM [29], as well as a synergistic postural adjustment during involuntary movement and during functional movements [30,31,32,33].

There are several methods and devices that address postural sensorimotor control to improve posture in different positions (supine, sitting and standing) and in functional activities. They all include sagittal alignment, axial self-elongation, and stability of the shoulder and pelvic girdles. Regarding neurosensory devices (bosu ball, oak wood surface, foam rubber mat, roller, etc.), these are used in progression from stable to unstable surfaces and from supine to standing positions. In standing, greater plantar cutaneous information enhances postural control and balance [34,35], in addition to the fact that the basal bioelectric activity of the PFM is also greater [27]. Although we found some studies on the effect of sensorimotor postural control using unstable surfaces and virtual games such as the Wii in female PFD [12,13], it is still a poorly studied area and lacks enough methodological quality.

Among these postural sensorimotor control methods is the Perineal Proprioceptive Postural Re-education also called 5P^®^ LOGSURF method. This method main purpose is to improve the muscle properties of striated PFM through reflex responses related to the sensory motor control of posture, body awareness and balance through exercises performed on an oak wood surface. This device has two different sides, a more stable convex surface and an unstable surface colloquially called “surfing surface”. This method is widely used in different European countries, although no study has been found showing its efficacy, not even its effects, of which only one non-conventional publication can be found in healthy women [6]. Therefore, the main objective of this study was to investigate the effect of a single 5P^®^ LOGSURF session on PFM tone and strength and resting anal tone as primary outcomes, as well as intrarectal pressure and deep abdominal muscle electrical activity as secondary outcomes in women with and without PDF. As secondary objective we aimed to determine the intra-session variability for all outcomes in both groups. 

## 2. Materials and Methods

This study was a cross-sectional observational preliminary study conducted from September 2019 to February 2020. This study (OE21/2017) was approved by the local Hospital’s Clinical Research Ethics Committee in Alcalá de Henares, Madrid, Spain. The Strengthening the Reporting of Observational Studies in Epidemiology (STROBE) recommendations were followed.

### 2.1. Participants

Premenopausal women who were referred to the Research Unit of the «BLINDED» by their general practitioner, urologist, gynecologist, or midwife to receive physiotherapy management of PFD signs or symptoms were invited to participate. The inclusion criteria were self-reported signs or symptoms of stress or mixed UI, AI, and/or gynecologist diagnosis of stage 1 or 2 of pelvic organ prolapse (POP), according to the POP-Quantification Scheme [36]. The exclusion criteria were aged ≤ 18 years, pregnancy, urinary tract infection diagnosis or symptoms, diagnosed neurological or musculoskeletal disorders, cognitive impairment, previous pelvic or urogynecology surgeries and conditions that could cause pain to the intracavitary insertion (perianal abscess, hemorrhoids, or perineal pain).

As a comparison non-PFD women 18–45 years old with no signs and symptoms of PFD, from «BLINDED» were recruited. Exclusion criteria were the same exclusion criteria for the PFD women.

All women provided their written informed consent before entering the study.

### 2.2. Measures

Demographic, identification, and clinical data were collected (L.F.-A) prior to single 5P^®^ LOGSURF session: Age, body mass index, education level, number of pregnancies, type of delivery, PFD symptoms and their severity, other pathologies, constipation, medication intake, regular physical activity practice, and prior postural training practice.

PFD symptoms and its impact on quality of life (QoL) were collected through the Spanish versions of the Pelvic Floor Disability Inventory (PFDI-20) and Pelvic Floor Impact Questionnaire (PFIQ-7) short forms. Both questionnaires are valid, reliable, and responsive for assessing symptom severity and impact on QoL in women with PFD [37,38]. 

A double balloon latex catheter for anorectal manometry (Rectalis, Phenix, Montpellier, France) was used to collect resting anal tone and intrarectal pressure. 

A commercially available instrumented plastic dynamometric speculum (Pelvimeter, Phenix, Montpellier, France) was used to measure PFM tone and PFM intravaginal peak force generated through MVC of the PFM. All dynamometry PFM tone numeric values are presented as raw values, that is, ambient pressure value included (170 gr). Both probes were used with protective latex or polyethylene covers.

Neuromuscular activity of the deep abdominal muscles (internal oblique and transversus abdominis muscles) was recorded using surface electromyography (Phenix Liberty, Vivaltis, Montpellier, France) with paired adhesive electrodes of 50 × 50 mm (Primtrode, Prim S.A, Madrid, Spain).

### 2.3. Procedure

Two physiotherapists (L.F.-A&B.A.-M) who had more than five years of experience in the physiotherapy management of PFD performed all the measurements. Firstly, L.F.-A explained the procedure in detail to the participants. Thereupon, B.A.-M performed the placing and checking of the measuring instruments. Women were instructed to situated lying on their left side with knee and hip flexion. In this position the anorectal catheter was inserted. Initially the rectal balloon (was inflated 5 mL and the anal balloon 2 mL. One minute after insertion, the rectal balloon was inflated 5 mL more and the anal balloon was inflated 2 mL more. Once the anorectal catheter was inserted, participants were instructed to stand up and once standing, the participant guided the closed, lubricated arms of the dynamometer into her mid-sagittal vaginal plane. L.F.-A held the dynamometer aligned comfortably into the participant’s vaginal canal throughout the experiment. Thirdly, adhesive electrodes were placed 2 cm caudal and anterior to the left and right anterior superior iliac spine. Before starting the single 5P^®^ LOGSURF session, PFM tone, resting anal tone, intrarectal pressure, and deep abdominal muscle activity at rest were recorded in standing position. Furthermore, a PFM MVC was requested in this position, asking to squeeze and lift the PFM in isolation as strong as possible; and a deep abdominal muscles contraction were also requested, asking to draw-in the inferior abdominal wall towards the spine without any lumbopelvic movement. 

All the participants performed a single 5P^®^ LOGSURF 30 min session. The 5P^®^ LOGSURF method is made up of three phases: A static (ST) phase that begins on the convex side (CS) of the oak wood surface (with a length of 9.5 cm, a width of 17 cm, a height of 7.5 cm and a weight of 2 Kg), more stable, in which balance must be maintained while standing. An upright posture should be maintained, keeping scapulae and pelvis aligned, the hips slightly internal rotated, the knees slightly bent, the feet parallel, the arms hang naturally down the sides of the body, and looking straight ahead. This phase lasts 20 min. In the second phase, also ST, the balance must be maintained in the same posture described above, but this time on the surf side (SS) for 5 min. Finally, in the third phase, dynamic (DIN) and also on the SS, the balance must be maintained by performing exercises for 5 min holding with arms extended an elastic resistance band (7 kg resistance, Nyamba model, Decathlon) attached to the oak wood surface. The upper limb exercises consist of slowly pulling the elastic resistance band to the top of the chest maintaining scapulae stabilized during prolonged exhalation. During inspiration, the arms slowly return to the starting position. This arms movement is repeated throughout the DIN phase (Figure 1). 

So, with the probes in, the participant was asked to climb on an oak wood surface. During the first phase on the CS for 20 min, a physiotherapist (L.F.-A) instructed the participant to correctly maintain the standing posture described above. In addition, PFM tone, resting anal tone, intrarectal pressure and deep abdominal muscles activity were recorded at different time points: At minutes 1 and 20 (1′ and 20′-CS). Next, the participants, without removing the probes at any time, came down from the CS to climb to the SS, and start the second and third phases for 10 min, 5 min for ST phase, and 5 min for DIN phase. PFM tone, resting anal tone, intrarectal pressure and deep abdominal muscles activity were also recorded at different time points: At minute 5 of the ST phase (5′-SS-ST) and at minute 5 of the DIN phase (5′-SS-DIN). In all phases a physiotherapist (L.F.-A) supervised that the posture was correct and, if necessary, provided indications for its correction.

Once single 5P^®^ LOGSURF 30 min session was finished, participants came down from the oak wood surface. PFM tone, resting anal tone, intrarectal pressure and deep abdominal muscle activity at rest were then recorded. A PFM MVC and a deep abdominal muscles MVC were also requested.

### 2.4. Data Analysis

Statistical analysis was conducted using SPSS Version 23.0 (IBM Corp, Armonk, NY, USA). Descriptive statistics were calculated using the arithmetic mean and SD as indices of central tendency and dispersion for the quantitative variables or using the median and interquartile ranges when wide dispersions conditioned the interpretation of the variable. The assumption of normality was verified by the Shapiro–Wilk statistical test. Absolute and relative percentage frequencies were used for the categorical variables. The inferential analysis was estimated with a 95% CI, considering a *p*-value < 0.05 as statistically significant. 

The measure of association between two categorical variables was carried out using Fisher’s exact test. Student’s t-Test (for independent samples) and paired Student’s t-Test (for dependent samples) were used in order to determine the association between a dichotomous independent variable and a quantitative dependent variable of parametric distribution. Mann–Whitney U Test was used for independent samples when the dependent variable violated the assumption of normality, or Wilcoxon Test for paired samples.

A repeated measures analysis of variance (ANOVA) was performed to investigate the effect of the within-group factor “time” (four categories: 1′-CS, 20′-CS, 5′-SS-ST, 5′-SS-DIN) and the between-groups factor (patients and controls) on those outcome variables with a normal distribution. When sphericity could not be assumed according to the results of Mauchly’s test, Greenhouse–Geisser correction was employed. For between-groups interaction analysis, it also was required a homogeneous covariance matrix. The effect size for each main effect and interaction in the ANOVA was measured with the partial eta squared (ηp2), with 0.01–0.059 representing a small effect, 0.06–0.139 a medium effect and >0.14 a large effect. For significant findings in the ANOVA, a post-hoc analysis for multiple comparisons was conducted using Bonferroni correction. We calculated the effect size (Cohen’s d) for normally distributed data, considering 0.20–0.49, 0.50–0.79 and >0.80 to be small, medium, and large effect sizes, respectively [39]. 

Friedman’s test was conducted for assessing intrasession changes in those variables where ANOVA conditions were violated. Post-hoc analysis was conducted using Wilcoxon test with Bonferroni correction.

## 3. Results

In total, 30 women were included in the study, 11 in the non-PFD group and 19 in the PDF group. All women completed the single 5P^®^ LOGSURF 30 min session (see flow diagram in Figure 2). 

### 3.1. Study Participants Characteristics

Thirty women aged between 24 and 52 years old participated in the study; 19 with PFD and 11 without PFD. Clinical and demographic characteristics are shown in Table 1. No participant reported medication intake.

### 3.2. Effect of Single 5P^®^ LOGSURF Session 

The comparative analysis between pre- and post- single 5P^®^ LOGSURF session related to PFM tone, resting anal tone, intrarectal pressure, deep abdominal tone, PFM contraction and deep abdominal contraction is shown in Table 2.

PFM tone and PFM MVC decreased in both groups after a single 5P^®^ LOGSURF session. PFM tone showed statistically significant differences (PFD group: *p* = 0.09, *d* = 0.72; non-PFD group: *p* = 0.003, *d* = 0.49) with medium and small effect sizes respectively in PFM tone. PFM MVC also showed statistically significant differences (PFD group: *p* = 0.016; non-PFD group: *p* = 0.005). When comparing pre-session values between both groups, statistically significant differences were also found (*p* = 0.003), with higher values in the non-PFD group.

Decreasing resting anal tone and intrarectal pressure was also observed after single 5P^®^ LOGSURF session in both groups, although there were not statistically significant differences.

Deep abdominal muscle MVC activity increased in both groups with statistically significant differences (PFD group: *p* < 0.001; non-PFD group: *p* = 0.003). 

### 3.3. 5P^®^ LOGSURF Session Variability 

#### 3.3.1. Session Variability on PFM Tone and Resting Anal Tone

Table 3 shows the descriptive data of primary and secondary outcome measures across the single 5P^®^ LOGSURF session. Changes and confidence intervals for PFM tone and resting anal tone are shown in Figure 3.

PFM tone decreased throughout single 5P^®^ LOGSURF in the PFD group, but no statistically significant differences were found (*F* = 3.04, *p* = 0.057, ηp2 = 0.144). Regarding non-PFD group PFM tone showed statistically significant changes with a large effect size throughout the session (*F* = 5.06, *p* = 0.006, ηp2 = 0.336). Post-hoc analysis revealed differences between 1′-CS and 5′-SS-ST, with a large effect size (*p* = 0.043, *d* = 1.04).

Concerning the inter group analysis, ANOVA showed no interaction of the group over the results (*F =* 2.70, *p* = 0.067, ηp2 = 0.088).

Resting anal tone increased throughout single 5P^®^ LOGSURF session with a large effect size in the PFD group (*F =* 8.99, *p* = 0.002, ηp2 = 0.333). Post-hoc analysis showed significant differences with medium effect size between 1′-CS and 5′-SS-DIN (*p* = 0.029, *d* = 0.74), and large effect sizes between 20′-CS and 5′-SS-DIN (*p* < 0.001, *d* = 1.18) and between 5′-SS-ST y 5′-SS-DIN (*p* < 0.001, *d* = 1.46). Statistically significant changes were not found in non-PDF group (*F =* 3.57, *p* = 0.063, ηp2 = 0.263). 

Inter group analysis revealed no effects of the group over the resting anal tone (*F:* 1.46, *p* = 0.242, ηp2 = 0.050).

#### 3.3.2. Session Variability on Intrarectal Pressure and Deep Abdominal Muscle Electrical Activity

Intrarectal pressure increased throughout single 5P^®^ LOGSURF session. Non-parametric analysis of secondary variables showed statistically significant changes in both groups (PFD group: *p* = 0.002; non-PFD group: *p* = 0.018). Post-hoc analysis revealed significant changes in both groups between 5′-SS-ST and 5′-SS-DIN (PFD group: *p* = 0.024; non-PFD group: *p* = 0.042), 

Deep abdominal muscle electrical activity increased throughout single 5P^®^ LOGSURF session, statistically significant changes were found in both groups (*p* < 0.001). Post-hoc analysis showed significant changes in both groups between 1′-CS and 5′-SS-DIN (PFD group: *p* = 0.03; non-PFD group *p* = 0.018), between 20′-CS y 5′-SS-DIN (PFD group: *p* < 0.01; non-PFD group: *p* = 0.024) and between 5′-SS-ST y 5′-SS-DIN (PFD group: *p* < 0.01; non-PFD group *p* = 0.018), with an increase in muscle activity.

No adverse effect was reported.

## 4. Discussion

To the best of our knowledge, this is the first study investigating the effects of a single 5P^®^ LOGSURF session in women with and without PFD. The findings from this study suggest that PFM tone and PFM MVC decreased in both groups after a single 5P^®^ LOGSURF session, with a medium effect size for women with PFD. Throughout the session, PFM tone also showed a tendency to decrease in both groups, although women without PFD recovered initial values in the last 5′ of the session, this means performing arm exercises with elastic resistance band. Furthermore, throughout these last 5′, intrarectal pressure and deep abdominal muscle activation increased in all women, and resting anal tone in only women with PFD. Nevertheless, once the single 5P^®^ LOGSURF was over intrarectal pressure and resting anal tone decreased in both groups, and deep abdominal muscle MVC increased. 

Our study included women with and without PFD. The average age of participants, around 31 years old in the group without PFD and 39 years old in the group with PFD, is lower than other studies including premenopausal women with and without PFD where ages were from 38 to 48 respectively. Previous studies have shown that PFM tone is significantly lower in women over 46 years old [40]. However, in our study there are no significant differences between groups in pre-session PFM tone although there are significant differences between the mean age of participants with and without PFD. Regarding the PFM basal tone and the PFM MVC, the data were representative of women with and without PFD [40]. Although most studies take these values in the lithotomy position, that is, in the supine position, and in the current study they have been taken in the standing position, the highest PFM resting tone found may be directly influenced by gravity. This fact agrees with Morgan et al. [41] who found that the vaginal resting pressure is significantly higher in the standing compared to the supine position. In any case, the difference shown by other studies regarding PFM resting tone and strength between healthy women and PFD women are similar to ours [40]. Regarding the resting anal tone, although non-PFD women had higher anal resting tone than PDF women, both groups show values within the range of normal female values described in the literature [42]. Concerning PFD symptoms, PFDI-20 and PFIQ-7 questionnaires showed very low scores (8.3 and 0 respectively), with values of 0 in all dimensions of both questionnaires, agree with other studies in women without PFD [43,44,45]. Nevertheless, women with PFD showed higher values than non-PFD women (PFDI-20: 22.9 and PFIQ-7: 19.1) which means that the severity of the symptoms was mild (the maximum score of both questionnaires is 300 points) compared to other studies in women with PFD (PFDI-20: From 55 to 21 points and PFIQ-7 from 39 to 63 points). However, these studies, as the present study, showed higher values in PFDI-20 than in PFIQ-7. The mild severity of PFD symptoms in the present study are consistent with the low risk factors shown in women with PFD, such as age (39.11), normal weight (24.68), parity (2.16) or stage 1 or 2 POP [46]. 

Regarding 5P^®^ LOGSURF method, the hypothesis that would explain its effect on PFM is to maintain the posture in an unstable plane. Several studies have analyzed certain postural sensorimotor control methods and their effect on the reflex activation of the PFM and deep abdominal muscles, due to influence of body positions on muscle recruitment. Specifically, some studies showed the influence of pelvic tilt, ankle position, foot position, or scapular position on PFM activation. Capson et al. showed the influence of different pelvic tilt positions on in PFM activation and vaginal pressure [30]. Different authors stated the relationship between ankle dorsiflexion and PFM activity in standing position [47,48]. In our study the women’s posture on the unstable plane was with neutral pelvic tilt, knees slightly flexed, and hips slightly internal rotated what could have activated the iliococcygeus muscle [49]. Scapular activation also played an important role. Hodges et al. described, in 7 healthy subjects, an activation of the PFM during the shoulder flexion movement [23], also performed by the women in our study throughout the DIN phase in the SS. 

Although practically the entire sample had previous experience with other postural methods, we can hypothesize that the postural work used in a single 5P^®^ LOGSURF session is not enough to provoke a response in the PFM.

In our study we found a decrease of the passive and active PFM components (PFM and MVC tone) over time. We hypothesized that at first women revealed some discomfort to maintain standing position on the oak, showing a greater initial tonic activation of the PFM, and after a while, when women were more comfortable on the oak tonic activation of the PFM was normalized, according to Czyrnyj et al. [50]. Furthermore, this could also be due to the role of musculoskeletal viscoelastic properties [51]. Tissues adapt to changes in their mechanical properties in response to load and sensory information provided during movement, prompting the central nervous system to adapt to changes. Variable loading can be effective in a few ways, one of which is that small variations in loading can promote a greater “mechanotransductive” effect through broader stimulation of the mechanoreceptors and prevention of accommodation; the phenomenon of accommodation is well known from a variety of other stimuli such as temperature, pressure, and light [52]. 

According to our results, PFM tone decreases in the 20 min in which women maintain posture in the CS and experiences a slight increase when varying in surface (SS) and load (DIN). Progressive overload is one of the training programs principles to improve strength and endurance [53]. This suppose that increasing the difficulty of a specific target when is achieved is needed to gain muscular force and resistance. Thus, the maintenance of a static standing posture over time without any challenge for the patient would not hypertrophy the muscle but would produce a neuromuscular accommodation. This means that the very time-consuming CS phase should be questioned. Other postural exercises such hypopressive exercises have shown an increase in the neuromuscular activity of the PFM, however these exercises did not only maintain a posture but added a respiratory maneuver [8].

In this study, the variation in neuromuscular activity of the deep abdominal muscles has been described. Transversus abdominis muscle has a key role in the regulation of intra-abdominal pressure and postural control in the standing position [54]. During resisted shoulder flexion, activation of the transversus abdominis muscle related to changes in the center of mass has been described in healthy men [54]. Our study showed significant increases in deep abdominal muscles activity throughout the session but without significant changes in the comparison before and after the 5P^®^ LOGSURF session. This could be because a single session was evaluated and because the activation of transversus abdominis muscle may not be so generic but may require specific asymmetric and rotational movements [55]. In the aforementioned study on hypopressive exercises, a significant increase in the deep abdominal muscles activity was found, also related not only to the posture used but also to the respiratory maneuver in expiratory apnea, where these muscles are expected to be activated [8]. After the session, deep abdominal muscles MVC was greater, which could be considered to improve the ability to voluntarily contract these muscles.

Regarding resting anal tone, our study showed a tendency to decrease throughout the single 5P^®^ LOGSURF session with no significant changes in both groups. These results contradict those of an unpublished study on the effect of a single 5P^®^ LOGSURF session which described an increase of resting anal tone immediately after the single 5P^®^ LOGSURF session [56]. Although this unpublished study does not provide enough information on the methodology and procedure, this difference could be due to the sample characteristics. In our study women with mild PFD symptoms were included. Their anal resting tone and intrarectal pressure were lower than healthy women but are within normal values [42]. The unpublished study included 29 women (mean 47 years old) with dyschezia and without active anal lesions. Dyschezia is a condition associated with chronic constipation and higher manometric anal pressures [56]. 

Finally, no intrarectal pressure relevant changes were obtained between pre- and post-session. During 20 min in which women maintained standing position the intrarectal pressure increased in the surf side (ST and DIN) in both groups. This may be due to the anticipatory muscular activation to achieve postural and balance adjustment [23]. On unstable positions abdominopelvic muscles are adapted to the demands to achieve these adjustments and may provoked an intrarectal pressure increase [24].

However, some limitations should be noted. First the small sample size due to the preliminary nature of the study and effect of one single session was analyzed and long-term effects were not studied. The measures presented are primarily descriptive. 

A prospective study could be taken up to evaluate the short and long-term effects of the 5P^®^ LOGSURF program by using a different target population such as women with different types of PFD or with higher symptoms or with comparisons of different other postural sensorimotor control programs.

## 5. Conclusions

These results suggest that 5P^®^ LOGSURF method described in this study may be interesting if is performed by women with mild symptoms of PFD or healthy women to achieve a decrease in PFM tone in women who manifested pain to intracavitary techniques or practices. Our findings show a loss of strength in PFM but an improvement in deep abdominal muscles voluntary contraction. These results denote 5P^®^ LOGSURF method combined with other more specific PFM techniques may be an interesting option to achieve a more comprehensive abdominopelvic treatment in certain pelvic floor conditions but not to increase the strength and tone of the PFM. Further research with higher sample sizes and population with higher PFD symptoms would help in generalizing the findings of this study.

## Figures and Tables

**Figure 1 ijerph-18-03708-f001:**
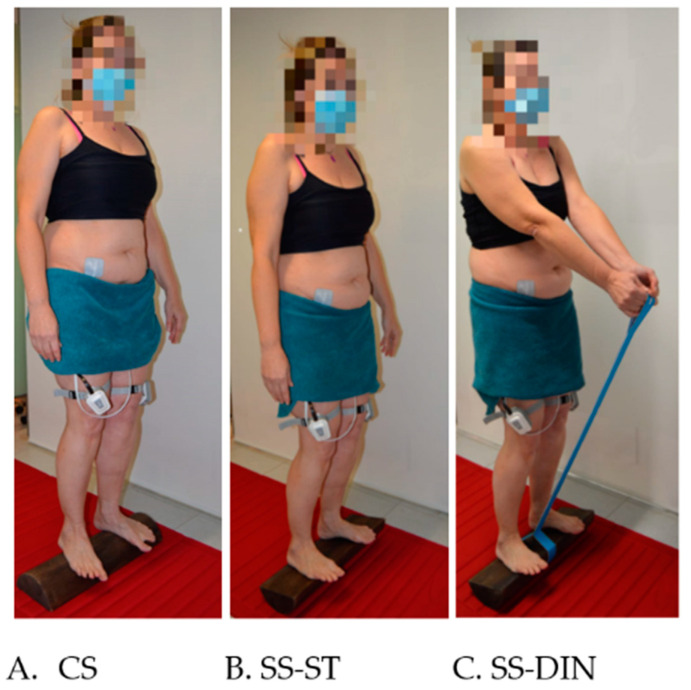
(**A**). Static phase on the convex side of the oak wood surface (CS); (**B**). Static phase on the surf side of the oak wood surface (SS-ST); (**C**). Dynamic phase on the surf side of the oak wood surface performing exercises with elastic resistance band (SS-DIN).

**Figure 2 ijerph-18-03708-f002:**
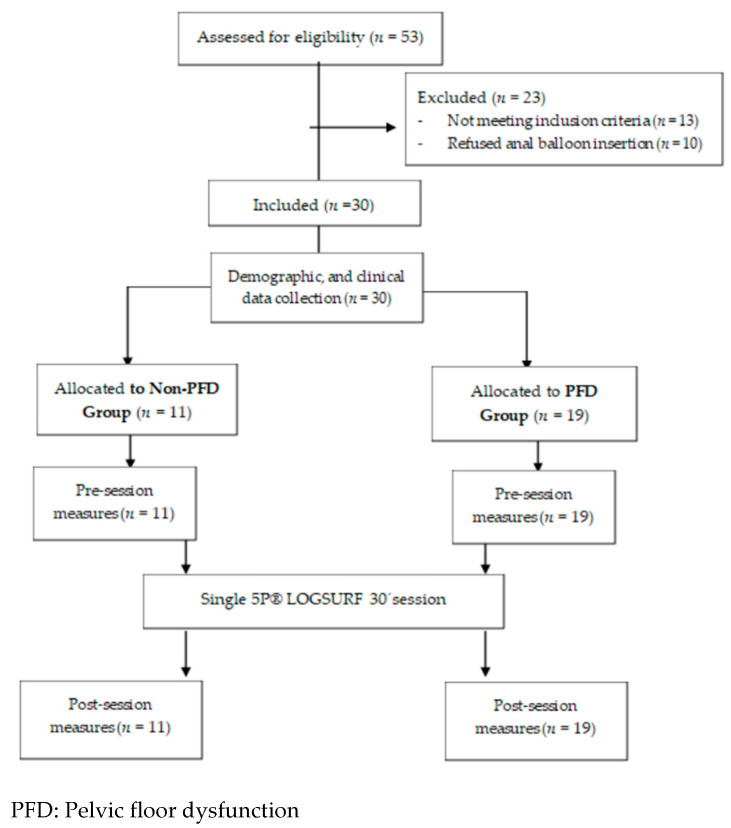
Flow diagram of participants.

**Figure 3 ijerph-18-03708-f003:**
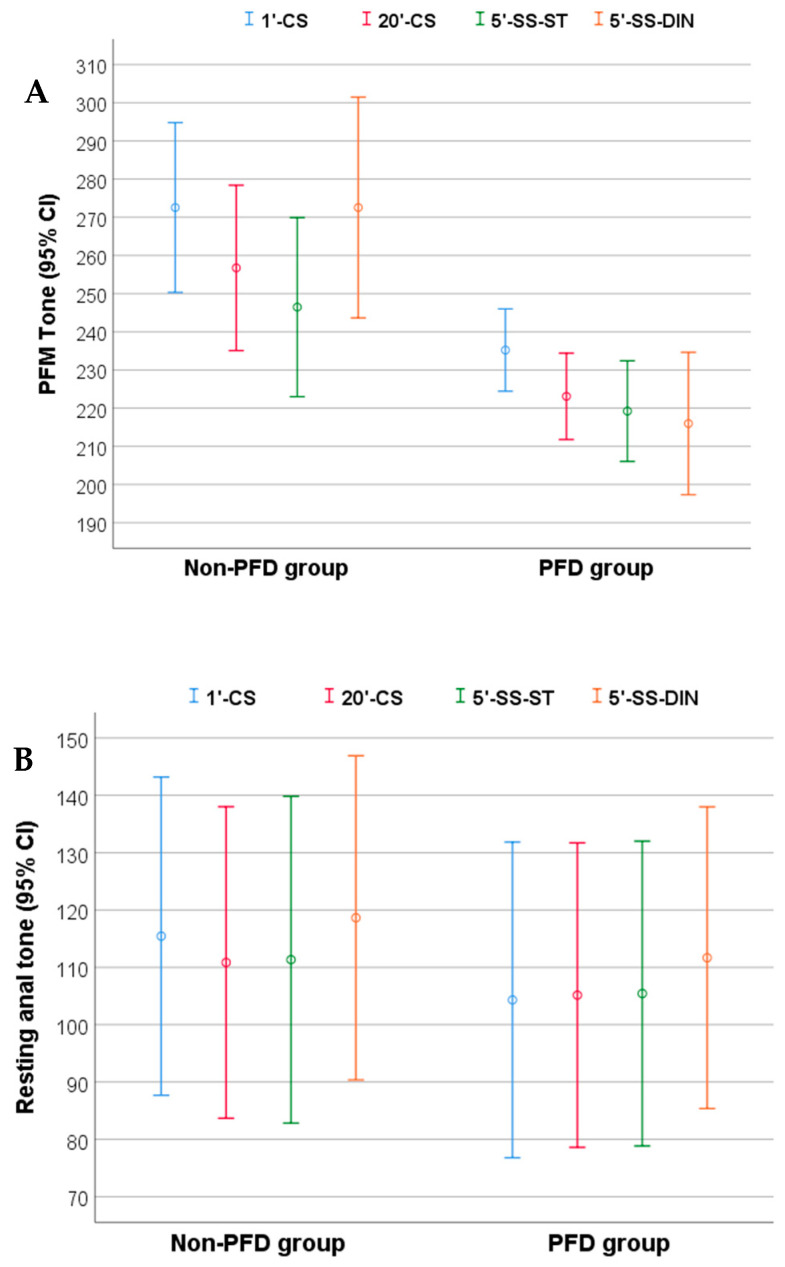
Pelvic floor muscle tone (**A**) and resting anal tone (**B**) throughout single 5P^®^ LOGSURF session. PFM: Pelvic floor muscle; PFD: Pelvic floor dysfunction; CS: Static phase on the convex side; SS-ST: Static phase on the surf side; SS-DIN: Dynamic phase on the surf side.

**Table 1 ijerph-18-03708-t001:** Demographic and clinical characteristics of participants. Values are numbers (percentages) unless stated otherwise.

	Non-PFD Group	PFD Group	*p*-Value
	(*n* = 11)	(*n* = 19)	
**Demographic and Anthropometric Data**			
Age (years), Χ (SD)	31.18 (6.11)	39.11 (6.20)	**0.002 *****
Education level			**0.000 ****
Elementary or professional education	0 (0)	4 (21.05)	
University Degree	11 (100)	15 (78.90)	
Body mass index (kg/m^2^), Χ (SD)	23.31 (2.95)	24.68 (4.93)	0.576 *
Parity, Χ (SD)	1.64 (2.15)	2.16 (1.21)	0.168 *
Type of delivery			
Caesarean	1 (9.1)	3 (15.80)	1 **
Vaginal	5 (45.50)	19 (100)	**0.001 ****
**Clinical Data**			
Urinary incontinence	-	13 (68.40)	-
SUI	-	9 (47.40)	-
UUI	-	2 (10.50)	-
MUI	-	2 (10.50)	-
Anal incontinence	-	9 (47.40)	-
PFDI-20 total, Median (IQR)	8.3 (20.8–0)	22.9 (50–16.7)	**0.011 ***
POPDI, Median (IQR)	0 (8.3–0)	8.3 (16.7–0)	0.063 *
CRADI, Median (IQR)	0 (13.1–0)	6.25 (15.6–3.1)	**0.011 ***
UDI, Median (IQR)	0 (8.3–0)	4.2 (25–0)	0.156 *
PFIQ-7 total, Median (IQR)	0 (4.8–0)	19.1 (33.3–4.8)	**0.001 ***
POPIQ, Median (IQR)	0 (0–0)	0 (14.3–0)	0.068 *
CRAIQ, Median (IQR)	0 (0–0)	0 (9.5–0)	0.081 *
UIQ, Median (IQR)	0 (0–0)	4.8 (19–0)	**0.014 ***
Concomitant diseases			
Anxiety	0 (0.00)	4 (21.10)	0.268 **
Other pathologies (scoliosis, auditive diseases, etc.)	1 (9.10)	5 (26.40)	0.372 **
Pads/day, Χ (SD)	-	0.16 (0.38)	-
Constipation	6 (54.50)	9 (47.40)	1 **
Regular physical activity	10 (90.90)	10 (52.60)	0.099 **
Physical activity (minutes/week), Median (IQR)	120 (60–240)	120 (90–150)	0.567 *
Postural training experience			
5P^®^ LOGSURF method	1 (9.10)	1 (5.30)	1 **
Other postural methods	10 (91)	18 (95)	1 **

SD: Standard deviation; Χ: Mean; IQR: Interquartile range; PFD: Pelvic floor dysfunction; SUI: Stress urinary incontinence; UUI: Urgency urinary incontinence; MUI: Mixed urinary incontinence; PFDI-20: Pelvic Floor. Distress Inventory Short Form; POPDI: Pelvic organ prolapse distress inventory; CRADI: Colorectal–anal distress inventory; UDI: Urinary distress inventory; PFIQ-7: Pelvic Floor Impact Questionnaire Short Form; POPIQ: Prolapse impact questionnaire; CRAIQ: Colorectal–anal impact questionnaire; UIQ: Urinary impact questionnaire. * U-Mann-Whitney test; ** Fisher´s test; *** Student´s *t*-test. The significant differences are highlighted in bold.

**Table 2 ijerph-18-03708-t002:** Average change for each outcome between pre- and post-single 5P^®^ session.

Outcomes	Group	Pre-5P^®^ Session	Post-5P^®^ Session	Pre-Post 5P^®^ Session Difference	Intragroup Pre- vs. Post-5P^®^ Session, *p*-Value	Between Groups Pre-Post 5P^®^ Session Difference, *p*-Value	Between Groups Pre-5P^®^ Session, *p*-Value
PFM Tone (g)	PFD	238.60 (18.33)	222.65 (24.89)	−15.95 (23.79)	0.009 ***	0.442 ****	0.052 ****
	Non-PFD	266.64 (40.96)	244.09 (27.58)	−22.55 (19.36)	0.003 ***		
PFM MVC (g)	PFD	284.23 (88.50)	249.45 (77.72)	−34.66 (49.45)	0.000 *	0.179 **	**0.003 ****
	Non-PFD	394.54 (90.45)	337.20 (79.34)	−52.60 (34.28)	**0.005 ***		
Resting anal tone (mmHg)	PFD	110.89 (57.99)	104.54 (54.57)	−6.35 (15.34)	0.088 ***	0.741 ****	0.848 ****
	Non-PFD	114.77(42.73)	106.59 (42.65)	−8.18 (12.72)	0.059 ***		
Intrarectal pressure (mmHg)	PFD	104.31 (34.18)	101,44 (37,76)	−2.87 (8.81)	0.372 *	0.667 **	0.863 **
	Non-PFD	111.88 (23.82)	101.65 (42.69)	−10.24 (22.62)	0.285 *		
Deep abdominal muscle activity (µV)	PFD	5.64 (7.62)	4.29 (2.11)	−1.34 (7.52)	0.753 *	0.596 **	0.288 **
	Non-PFD	5.55 (3.55)	6.55 (5.00)	1 (2.48)	0.248 *		
Deep abdominal muscle activity MVC (µV)	PFD	17.24 (14.10)	20.34 (13.73)	1.85 (7.70)	**0.000 ***	0.853 **	0.143 **
	Non-PFD	26.77 (17.15)	30.20 (18.34)	0.56 (10.28)	**0.003 ***		

PFM: Pelvic floor muscles; PFD: Pelvic floor dysfunction; MVC: Maximal voluntary contraction; g: Grams; mmHg; millimeters of mercury; µV: Microvolts. * Wilcoxon test; ** U-Mann–Whitney test; *** Paired Student *t*-test; **** Student *t*-test. The significant differences are highlighted in bold.

**Table 3 ijerph-18-03708-t003:** Intragroup differences in pelvic floor muscles tone and resting anal tone across a single 5P^®^ LOGSURF session.

		1′-CS	20′-CS	5′-SS-ST	5′-SS-DIN
PFM Tone (g). Χ (SD)	PFD	235.21	223.09	219.23	215.98
		(22.35)	(23.48)	(27.34)	(38.69)
	Non-PFD	272.55	256.78	246.45	272.54
		(33.10)	(32.23)	(34.92)	(43.02)
Resting anal tone (mmHg). Χ (SD)	PFD	104.32	105.15	105.42	111.67
		(57.12)	(55.09)	(55.14)	(54.58)
	Non-PFD	115.44	110.85	111.34	118.64
		(41.32)	(40.45)	(42.42)	(42.07)
Intrarectal pressure (mmHg). Median (IQR)	PFD	117.90	117.20	114.60	121.20
		(134.50–117.90)	(130.10–86.90)	(128.10–80.30)	(135.10–81.40)
	Non-PFD	122.30	122.20	125.60	128.90
		(125.60–82.30)	(131.10–75.60)	(126.70–67.90)	(130.20–90.00)
Deep abdominal muscle activity (µV). Median (IQR)	PFD	4.50	4.50	4.50	8.00
		(6.00–3.50)	(6.00–3.50)	(7.00–3.50)	(10.00–6.00)
	Non-PFD	5.00	6.00	5.00	9.50
		(15.00–4.00)	(12.00–3.00)	(18.00–3.00)	(23.00–6.00)

Χ: Mean; SD: Standard deviation; IQR: Interquartile range; PFD: Pelvic floor dysfunction; µV: Microvolts; mmHg; millimeters of mercury; CS: Static phase on the convex side; SS-ST: Static phase on the surf side; SS-DIN: Dynamic phase on the surf side.

## Data Availability

The data presented in this study are available from the corresponding author upon reasonable request.

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
