# Peer review of "Postural Sensorimotor Control on Anorectal Pressures and Pelvic Floor Muscle Tone and Strength: Effects of a Single 5P^®^ LOGSURF Session. A Cross-Sectional Preliminary Study"

_ijerph, 2021, doi:10.3390/ijerph18073708_

Round 1

Reviewer 1 Report

This important and interesting manuscript: "Postural sensorimotor control on anorectal pressures and pelvic 2 floor muscle tone and strength: effects of a single 5P® LOG- 3 SURF session. A cross-sectional preliminary study", investigated the effects of a single 5P ® LOGSURF session in women with and without PFD and suggest that PFM tone and PFM MVC decreased in both groups after a single 5P ® LOG-  SURF session, with a medium effect size for women with PFD.

In page 3 the authors demonstrated the inclusion and exclusion criteria. Please clarify, what about women treating with different medicine treatment such as antidepressants or pain relief s or birth control pills and more? In addition, what about a woman who has had pelvic surgery or urogynecological surgery?

Measures- Please explain about the double balloon latex test, and explain exactly when it was done during the experiment?

Conclusion should be taken carefuly as this is a small sample and the methods should be clarified.

Reviewer 2 Report

The authors paid special attention to the effects of 5P®LOGSURF session for pelvic floor muscle (PFM) tone and strength (MVC), resting anal tone, intrarectal pressure, and deep muscles activation. In this study, such parameters were evaluated in 30 women including 11 without pelvic floor dysfunction (PFD) and 19 with PFD. As results, PFM tone and PFM MVC were decreased by a single 5P®LOGSURF session. On the other hand, deep abdominal muscle MVC was increased by the method. Finally, they concluded that 5P®LOGSURF method is useful for a variety of women with PFD.

I think that their study has important results to discuss the treatment strategies of women with PFD. In fact, I have no major question about methods and results. However, I have several questions before publication.

(Minor)

  1. You showed the information on concomitant diseases in Table 1. If you have any information on a medicine one takes constantly, please show it in 3.1. Study participants characteristics section.

  1. I think the age of participants in your study population is relatively low. In this study, exclusion criteria about the age is only ≤ 18 years. I would like to know the reason why elderly women is rare in this study population.
